# Kinematic Strategies for Sustainable Well-Being in Aging Adults Influenced by Footwear and Ground Surface

**DOI:** 10.3390/healthcare10122468

**Published:** 2022-12-07

**Authors:** Miao Tian, Ye Lei, Yunyi Wang, Shitan Wang, Jun Li, Shu Yuan

**Affiliations:** 1Key Laboratory of Clothing Design and Technology, Donghua University, Ministry of Education, Shanghai 200051, China; 2College of Fashion and Design, Donghua University, Shanghai 200051, China; 3Shanghai Yangzhi Rehabilitation Hospital (Shanghai Sunshine Rehabilitation Center), School of Medicine, Tongji University, Shanghai 201619, China

**Keywords:** kinematic strategies, aging adults, range of motion, footwear

## Abstract

Falls are an inescapable problem influencing the health and threatening the safety of older adults. Exploring the kinematic strategies of aging adults can help reduce the risk of falls. To study kinematic strategies of aging adults in response to footwear (flat shoes, toe spring shoes, rocker sole shoes) and ground surfaces (level ground, grassland and rock road), a 3D motion capture system and subjective stability evaluation, with 14 female participants, were performed. Results indicated that footwear and ground surfaces significantly impacted joint dynamics during walking. Compared with young adults, aging adults tended to adopt a more conservative walking pattern. Wearing different shoes on the three ground surfaces mainly reduced the ROM (range of motion) of the ankle (*p* < 0.05). By analyzing the objective and subjective results, rocker sole shoes gave aging adults a stronger sense of instability, so they controlled the movement of ankle joint initiatively. When walking on grassland and rock road, aging adults adjusted the movements of hip, knee and ankle joints to maintain gait stability. Aging adults are recommended to strengthen flexibility training of the ankle joint, perform hip adduction and abduction exercises, and wear rocker sole shoes to improve their balance ability and sustainable well-being.

## 1. Introduction

Falls are a significant public health concern given our world’s aging population [1]. In the United States, falls occur in more than 30% of older adults in the community [2,3] and are the leading cause of traumatic injury-related death, which is a neglected public health problem among populations, especially in the older adults [4,5]. An unsteady gait is one of the identifiable risk factors leading to the risk of falls. Injuries because of falls often remained stable or increased with age [6]. The consequences of falls in aging adults could be fractures, cardiovascular and cerebrovascular accidents, and even death, which will also decline the body’s ability to exercise, affect the quality of life and increase the social, medical, and economic burden [7,8].

In daily work, people may walk over various terrains with different surface characteristics [9,10]. Results indicated that more than 50% of falls in older adults occur outside the home and are mainly precipitated by uneven surfaces (47.6%) [5] because the individuals do not, or cannot, adapt their gait appropriately [11]. In older people, loss of balance is one of the significant contributors to falls [10,12,13]. Uneven surfaces lead to modifications in kinematic and stability characteristics of gait in aging adults. Older people usually have more difficulty than younger adults in maintaining balance when walking on irregular or uneven surfaces, such as wooden blocks randomly placed beneath a malleable surface, destabilizing rock surface, and uneven brick surface [1,4,10]. A more conservative walking pattern characterized by a slower walking velocity and shorter steps was preferred to be adopted [14,15,16]. Investigating the change of gait parameters on uneven surfaces is beneficial for determining the factors limiting safe locomotion and exploring appropriate methods to decrease the adverse effects of these challenging surfaces [14].

Footwear is an essential medium between the foot and walking surface, probably influencing balance control, and the risk of experiencing slips and trips [10,13]. Effects of age-related sensory degradation on foot position awareness and perception of floor slipperiness were investigated [17,18]. Heel height, heel geometry, and whether to wear shoes were significantly associated with an increased falls risk [15,17,19]. However, only a few studies have investigated the effects of shoe characteristics on gait stability in older people, focusing on either heel height or heel geometry [13,20,21,22,23]. Studies of heel height are carried out on young adults, older adults, and people who have worn high heels for a long time. Inconsistent selection of shoes between experiments led to inconsistent research conclusions. In general, wearing a higher heel will reduce gait stability. Research on older women pointed out that a heel height higher than 2.5 cm will nearly double the risk of falling. Therefore, it is recommended that the heel height of the elderly be less than 2.5 cm [24]. In addition to the change in heel thickness, the change in heel geometry also includes heel abduction, beveled heel shoe, and flared sole shoe design [20]. Studies have been conducted on the influence of abduction heels on the human body’s balance stability, and it has been found that this type of heel can reduce slippage in the shoe [25]. The front rocker refers to the front rocker angle designed according to the natural rocker degree of the forefoot (approximately 15°), which is called a toe spring shoe. Shoes with both front and back stilts are also called rocker sole shoes, which Masai Barefoot Technology first developed. Wearing rocker sole shoes, Nigg et al. [26] found that the ankle and knee joint activity decreased after wearing it, while Cox et al. [27] found that the ankle joint activity increased. Most of the research on toe spring shoes and rocker sole shoes is carried out for young adults.

The maintenance of human gait stability is related to the internal physiological balance system of the body, and is reflected through the locomotor system. The relevant research usually adopts the biomechanical methods of kinematics, dynamics, and physiology corresponding to the joint activities, skeleton, and muscle of the movement system. The angle of limb activity is an essential indicator for assessing the stability of the dynamic human balance. The principle of evaluating stability is to judge the body’s movement strategy by the following: analysis of ROM of human joints, significantly lower limb joints (hips, knees, ankles) in three cross-sections in the sagittal, coronal, and horizontal planes during the gait cycle under the action of different external environments; then judging the changes in the body’s morphology in the anterior–posterior, internal–external and up–down directions, and thus assessing the body’s gait stability [28]. Three-dimensional motion capture technology can be used to collect and process joint motions.

Understanding the kinematic strategies in aging adults when walking on uneven surfaces is significant for safety. To avoid the potential risks and safety issues of aging adults participating in the research, an age simulation suit used to demonstrate specific physical impairments of older age (e.g., strength and sensory losses) was used [29]. Footwear is the key element between the human foot and the road, and old adults will face different kinds of ground surface. The effects of footwear and ground surfaces on the changes in ROM of the lower limb in each plane were investigated, in order to explore the influence mechanism on gait stability.

## 2. Methodology

Considering the potential risks to older adults if participating in the unstable walking protocol, we adopted the age simulation suit, and recruited young adults to simulate the gait of older adults. This section introduced the age simulation suit, and the experimental design of the footwear and ground surfaces.

### 2.1. Age Simulation Suit

Compared with young adults, older adults are more likely to experience gait instability and trigger falls in daily life [30,31], so their gait stability issues are the primary consideration. There are certain safety risks with aging adults participating in related studies. Approaches have been adopted in previous studies to prevent potential risks through the addition of safety equipment, for example, lumbar protective gear. Still, this method may affect the subjects and experimental results, partly because the subjects have a leading consciousness after wearing the protective gear [32]. Some other studies have directly selected younger subjects to participate in the experiment instead of older adults, and have not demonstrated the feasibility of this alternative, raising questions about the applicability of the study’s findings in the aging population [33]. This study introduces an age simulation suit to make the young subjects’ characteristics more similar to those of the elderly, in physiology and behavior, to avoid the potential risks caused by directly selecting the elderly subjects, and improve the reliability of the research findings.

The age simulation suit for subjects used in this research was developed by Sanwa Manufacturing Company (Tokyo, Japan) [34]. By changing the position of the wearer’s center of gravity and the stability of the support surface, the device mechanically simulates the aging of the physiological balance system in older adults. The degradation of the human body’s perception system and movement system is simulated biomechanically through the force applied to the human body. Based on this, the component that simulates the degradation of the motion system is called the motion system simulation component, and the component that simulates the degradation of the perception system is called the simulation component of the perception system.

The simulation principle of the age simulation suit is shown in Figure 1. The motion system simulation components mainly include restraint belts and joint weight-bearing sandbags. Joints and muscles are an essential part of the body’s internal balance of the physiological system’s movement system. The internal balance system of the human body includes the nervous system, the sensory system, and the movement system, which are connected. In addition, finally, the movement system shows the human body’s balance. The elderly mobility simulation device can simulate the decline of the motion system and other related systems through restraint straps, joint weight-bearing sandbags, and other components, and then simulate the decline of balance ability. Therefore, this study mainly uses the effects of the motion system components in this set of equipment.

To analyze, quantitatively, its simulation effects of balance ability for old adults in the studies on gait stability, twenty-one participants (14 younger and 7 older female workers) were recruited (Figure 2). The figure displays the components of the age simulation suit used in this study. The participants, including a younger adult, a younger adult with the age simulation suit, and an older adult were also showed in this figure. A plantar pressure and gait analysis system was applied to obtain the posture balance and gait data of younger adults, younger adults with the age simulation suit, and older adults. The paired sample t-test was used to analyze the balance behavior of younger participants with and without the age simulation suit. The independent sample t-test was adopted to compare the results of younger participants with the age simulation suit, and older participants. Results indicated that the age simulation suit was able to simulate the decline of human motion and the perception system, and reduce the postural balance and gait stability of the human body. Younger adults with age simulation suits can simulate the postural balance performance, and reflect the balance behavior, of older adults during walking [34]. They will participate in the following study to avoid the safety risks to older adults.

### 2.2. Subjects

Fourteen young female adults (age: 23 ± 3; weight: 50 ± 5 kg; height: 160 ± 2 cm; BMI: 18.5–25) volunteered to participate in this study. They could move independently without relying on a walker to complete the gait test. There were no diseases among volunteers that affected stability, such as neurological diseases, fractures, and muscle injuries. All subjects provided written informed consent before participation in this study. The experimental protocol and procedure were approved by Donghua University. The subjects wore the age simulation suits when participating in the experiments.

### 2.3. Footwear and Ground Surface

Based on the results of literature research and survey interviews, the sole geometry was selected as the variable factor of the footwear. Flat shoes, toe spring shoes and rocker sole shoes were selected, as shown in Table 1. Three common pavement factor variables of level ground, grassland, and rock road are determined (Figure 3).

### 2.4. Experimental Protocol

The experiments included objective tests using motion capture equipment and subjective evaluation with the Likert Scale [35]. To explore the influence of footwear and ground surface factors on human gait stability, the ROM of the hip, knee, and ankle joints of the lower limbs were extracted by an Xsens MVN inertial motion capture system. The changes in the sagittal plane, the coronal plane, and the transverse plane were analyzed. Seven levels: very shaky (−3), shaky (−2), relatively shaky (−1), normal (0), relatively stable (1), stable (2), and very stable (3) were identified by the Likert scale that the participants could mark in a questionnaire according to their feelings.

The subjects were asked to wear the age simulation suit and experimental shoes to walk on the simulated road, including level ground, grassland, and rock road (7.5 × 0.5 m^2^) at a self-selected speed. A 1 m horizontal walkway is in front of each capture area, allowing the subject to reach a stable speed before data collection [10]. The lower limb joints’ motion parameters were recorded, and a subjective questionnaire survey was conducted during the tests. The test sequence was random, and the experiment was repeated three times for each sample combination.

### 2.5. Data Analysis

The Python language was applied to extract the joint angle data and determine the gait cycle by the maximum hip flexion touching the ground. Four steps (two steps on the left and two steps on the right foot) representing normal walking are selected for data analysis. The 12 calculated kinematics parameters are the ROM of the left and right hips, knees, ankles, and football in the coronal, transverse, and sagittal planes during a gait cycle. The effects of footwear, ground surfaces, and their interactions were analyzed using multivariate analysis of variance, with a significance level of 0.05. After the significance was confirmed, the Tukey post-test of the pairwise comparison was applied to different shoes, roads, and interaction treatments.

## 3. Results

### 3.1. ROM in the Sagittal Plane

Figure 4 shows the lower limb range of motion in the sagittal plane, when wearing the three types of shoes for the three ground surfaces. The ROM at the hip, knee, and ankle joint displayed a decreasing trend from flat shoes, toe spring shoes, to rocker sole shoes when walking on all three ground surfaces. The most significant decrease occurred in the ankle joint. When wearing different shoes, the ROM of the ball of the foot was related to the ground surfaces.

A multivariate analysis of variance (MANOVA) was performed on the differences in mean lower limb joints ROM in the sagittal plane under different ground surfaces. Table 2 shows significant differences between the three ground surfaces.

Varying footwear conditions under the level ground surface significantly impacted the ROM of the flexion–extension at the knee joint, and the dorsiflexion–plantarflexion at the ankle joint (*p* < 0.001). No significance was found for the flexion–extension movement of the hip and the ball of the foot (*p* = 0.508, *p* = 0.684). Results of multiple comparisons within groups for ROM in the sagittal plane of knee and ankle joints indicated that the flexion–extension at the knee joint was significantly higher in flat shoes (*p* = 0.003) and toe spring shoes (*p* = 0.040), than in rocker sole shoes. The dorsiflexion–plantarflexion at the ankle joint was significantly greater in flat shoes than in rocker sole shoes (*p* = 0.008).

Using the MANOVA to analyze the impact of wearing different shoes under grassland conditions, it was found that there were significant differences in the sagittal ROM of the knee and ankle joints (*p* = 0.009, *p* < 0.001). The results of multiple comparisons within the group found that compared with wearing flat shoes, the subjects’ flexion–extension at the knee joint was significantly smaller when wearing rocker sole shoes (*p* = 0.007).

A significant relationship between flat shoes, toe spring shoes, and rocker sole shoes (*p* < 0.001, *p* = 0.017) was observed for the ankle joint. The results of multiple comparisons within the group showed that the sagittal ROM of knee and ankle joints were consistent. When wearing flat shoes this ROM was significantly greater than for rocker sole shoes (*p* = 0.004, *p* = 0.002).

The investigation of the ROM of each joint in the sagittal plane demonstrated that shoe differences had a consistent effect pattern on the hip, knee, and ankle joints, while it varied for the ball of the foot depending on the ground surface. The ROM at the ankle joint was smaller in all three ground surface conditions in rocker sole shoes, compared to flat shoes (level ground, *p* = 0.008; grassland, *p* < 0.001; rock road, *p* = 0.002). Lv [36] reported that young subjects walking on level ground in rocker sole shoes tended to increase the dorsiflexion–plantarflexion at the ankle joint to maintain gait stability, compared with flat shoes. The reason for this may be the rocker sole design. When wearing rocking shoes, the moment of landing will increase the dorsiflexion angle between the ankle joint, and the ball of the foot. During walking, the increscent ankle plantarflexion will raise the ankle joints’ ROM in the sagittal plane [37,38]. However, different results were found in this study. The perceptibility of the participants may decrease after wearing the age simulation suit [39]. They could not change their gait to adapt to the instability caused by the rocker sole shoes. In addition, aging adults may adopt a more conservative kinematic strategy than younger subjects to cope with the sense of instability. Compared to flat shoes, the ROM of the knee joint in the sagittal plane was significantly smaller when wearing rocker sole shoes (level ground, *p* = 0.003; grassland, *p* = 0.007; rock road, *p* = 0.004), which was consistent with the ankle joint. This result also differs from the findings for younger subjects. Previous studies indicated that younger subjects had significantly greater ROM at the knee joint in the sagittal, when wearing rocker sole shoes rather than flat shoes. The ROM of flexion–extension at the knee changed the most during walking, producing a maximum flexion angle in the support phase later. Moreover, the rocker sole design increased the knee flexion angle, increasing the subject’s pedal extension force [36]. However, the participants in this study were less likely to make knee flexion adjustments, thus exhibiting smaller ROM of the flexion–extension at the knee joint when wearing rocker sole shoes. The effects of footwear on the ROM varied in the sagittal plane during walking on different ground surfaces (Figure 5).

Compared to the toe spring shoe, a significantly smaller ROM at the knee joint in the sagittal plane was found in the ground level with the rocker sole shoe. A significantly smaller ROM at the ankle joint in the sagittal plane was found in the grassland condition, when comparing the toe spring shoe to the rocker sole shoe. Although the toe spring shoe and the rocker sole shoe had a similar front stilt structure, the overall structure was different. Wearing the toe spring shoe can lead the foot to move forward, and avoid excessive medial–lateral deflection. The contact area with the ground was also increased to improve stability. In comparison, the human body will unconsciously sway back and forth and roll forward when wearing the rocker sole shoe [36]. For different surfaces, the participants would use different joint flexion and extension to maintain gait stability. Due to the unevenness of the grassland, the ROM of the ankle joint in the sagittal plane was more carefully controlled when wearing rocker sole shoes, compared to toe spring shoes.

There was no interaction between shoe and ground surface conditions on the ROM in the sagittal plane, shown by the MANOVA (hip, *p* = 0.962; knee, *p* = 0.971; ankle, *p* = 0.855; the ball of the foot, *p* = 0.791). However, there were significant differences between the ground surface conditions on the hip and knee joints in the sagittal plane (hip, *p =* 0.002; knee, *p* = 0.027). Multiple comparisons within groups revealed that the flexion–extension movement of the hip was significantly greater in the grassland conditions, compared to the level ground and rock road conditions (level ground, *p* = 0.002; rock road, *p* = 0.042). For the flexion–extension movement of the knee joint, this was much more significant when walking in the grassland, compared to the level ground (*p* = 0.021). The participants regulated their posture by increasing the hip and knee joint ROM in the sagittal plane when walking on grassland, compared to level ground. The flexion–extension movement of the hip was increased when walking on the grassland, compared to the rock road.

### 3.2. ROM in the Coronal Plane

The effect of footwear on the ROM of the hip, knee, ankle, and ball of the foot joints in the coronal plane showed different trends among ground surfaces. Figure 5a depicts that there is no significant change in the ROM of the hip on the level ground when wearing three types of footwear. The difference between toe spring shoes and flat shoes in grassland conditions was not significant, and smaller ROM when wearing rocker sole shoes was observed. In the rock road condition, the abduction–adduction at the hip joint showed a decreasing trend among flat shoes, toe spring shoes, and rocker sole shoes.

Figure 5b displays that the toe spring shoes led to the largest abduction–adduction at the knee joint among footwear in level ground and grassland conditions, followed by flat shoes and rocker sole shoes. The largest and smallest ROM was found when wearing flat shoes and rocker sole shoes, respectively; they were the smallest under the rock road condition. The rocker sole design aimed to reduce the knees’ ROM in the sagittal plane, and thereby reduce ROM in the coronal plane. In addition, ROM decreased most significantly under grass conditions.

Figure 5c illustrates that the ankle ROM, in the coronal plane in rocker sole shoes, were the smallest for all three footwear factors under the three road conditions. The participants had the smallest ROM in ankle eversion–inversion. 

As shown in Figure 5d, walking on level ground has the largest ROM in the coronal plane, compared to flat shoes and toe spring shoes for the ball of the foot. When walking on grassland and rock road, the ROM when wearing the rocker sole shoes were close to that of flat shoes, and toe spring shoes led to the smallest ROM in the ball of the foot.

The effect of footwear differences on the change in the coronal plane of joints walking on three types of ground surfaces was analyzed by MANOVA (Table 3). No significant effect of footwear was found when walking on level ground and rock roads.

There was a significant effect of footwear on the ROM of ankle joints in the coronal plane only when subjects were walking on the grassland (*p* = 0.030). Multiple comparisons within groups for the ROM of ankle joints in the coronal plane indicated that the eversion–inversion at the ankle, when wearing the rocker sole shoes, was significantly less than when wearing flat shoes (*p* = 0.048).

The effects of footwear on the ROM at the hip, knee, ankle, and ball of the foot varied among ground surfaces in the coronal plane. Combined with the results of the MANOVA, the ROM of joints in the coronal plane were significantly smaller when wearing rocker sole shoes, than when wearing flat shoes in grassland conditions. (*p* = 0.048). By observing the change in the effect on humans in its sagittal plane, it is not easy to maintain the dorsiflexion–plantarflexion of joints when wearing the rocker sole shoes. Similarly, this required gait balance control of the eversion–inversion of the ankle joint. In young adults, the inversion of the ankle joint during the corresponding gait cycle was greater than in flat shoes, while the corresponding ROM of joints in the coronal plane were smaller than in flat shoes [36]. Human movement in the coronal plane of the joints represented the joint motion in the left and right directions. Previous study indicated that there was a smaller range of movement of eversion and inversion when wearing rocker sole shoes. A greater angle of inversion meant a greater reduction in the angle of eversion to maintain the stability of aging adults.

There was no interaction between footwear and ground surfaces on the ROM change in the coronal plane (hip, *p* = 0.883; knee, *p* = 0.965; ankle, *p* = 0.834; the ball of the foot, *p* = 0.561). However, significant differences were found between ground surfaces for the ankle joint movement (*p* < 0.001). Multiple comparisons within groups revealed that grassland was significantly more significant than the eversion–inversion of the ankle, in the level ground and rock road conditions (level ground, *p* = 0.005; rock road, *p* < 0.001). The participants modulated their body posture by increasing the eversion–inversion of the ankle while walking on grassland, compared to the level ground and rock road conditions.

### 3.3. ROM in the Transverse Plane

The effect of footwear on the ROM of joints in the transverse plane varies between ground surfaces, except at the ankle joint. Figure 6a displays that under level ground conditions, the ROM of the hip joint in the transverse plane was the greatest when wearing toe spring shoes, compared to flat shoes and rocker sole shoes. In the grassland and rock road conditions, the intrarotation–extrarotation of the hip joint was the largest when wearing flat shoes, compared to toe spring shoes and rocker sole shoes.

Figure 6b shows that the effect of footwear on the knee was consistent among the three ground surfaces. The ROM in the transverse plane was the smallest when wearing rocker sole shoes, compared to flat shoes and toe spring shoes, and the difference between flat shoes and toe spring shoes was not significant.

Figure 6c depicts that the effect of footwear is the same for all three ground surfaces, with the greatest ROM with flat shoes, followed by toe spring shoes, and rocker sole shoes.

Figure 6d indicates that the effect of footwear in the ball of the foot differs by ground surfaces. In the level ground condition, the ROM in the transverse plane was greatest in rocker sole shoes, compared to flat shoes and toe spring shoes. In grassland conditions, the ROM with rocker sole shoes remained the greatest, followed by flat shoes, and toe spring shoes. Under gravel conditions, flat shoes displayed the most significant effect on the ball of the foot, followed by rocker sole shoes and toe spring shoes.

A MANOVA was performed on the effect of footwear differences on the ROM of joints in the transverse plane when walking on different ground surfaces (Table 4). The results of walking on level ground, grassland, and rock road indicated that the ROM of intrarotation and extrarotation of both ankle joints differed significantly between footwear types (*p* < 0.001). In contrast, the hip, knee, and ball of the foot did not differ significantly.

Multiple comparisons within the flat group showed significant differences in the ROM of joints in the transverse plane between the two, when wearing flat shoes, toe spring shoes, and rocker sole shoes. Multiple comparisons within the grassland group showed significant differences between flat shoes and toe spring shoes (*p* = 0.006), flat shoes and toe spring shoes (*p* < 0.001), and toe spring shoes and rocker sole shoes (*p* = 0.021). Multiple comparisons within the rock road conditions indicated significant differences between two of the three footwear types.

The results of joint movement in the transverse plane demonstrated that the effect of footwear differences on the hip, knee, ankle, and ball of the foot varied among ground surfaces. Combining the results of the MANOVA, wearing rocker sole shoes resulted in the smallest ROM of the ankle joint in the transverse plane, followed by toe spring shoes, and flat shoes had the largest (*p* < 0.001). Compared with flat shoes, participants wearing rocker sole shoes use ankle joint inversion and intrarotation increase to maintain gait balance. The grip of the ball of the foot increases the range of ankle joint inversion and intrarotation. Correspondingly, ankle eversion and extrarotation decrease. The ROM of the ankle is reduced, which may increase the risk of ankle sprains during walking [40].

The results of the multivariate ANOVA demonstrated no interaction between footwear and ground surfaces on the ROM of joints in the transverse plane (hip, *p* = 0.908; knee, *p* = 0.980; ankle, *p* = 0.934; the ball of the foot, *p* = 0.754). However, footwear significantly affected the intrarotation–extrarotation at the ankle joint (*p* < 0.001). Multiple comparisons within groups revealed that for participants walking on grassland and rock road, this was significantly smaller than on the level ground (*p* = 0.01, *p* < 0.001). The posture was regulated by reducing the ROM of the ankle joint when walking on grassland and rock road, compared to the level ground.

### 3.4. Subjective Evaluation of Stability

The results of the subjective stability scores of the participants, under different footwear and ground surfaces, were analyzed to explore their psychological characteristics when using kinematic strategies to maintain gait stability. Based on the subjective stability rating scale, a score of 0 or above indicates that the subject feels more stable during walking, while a score of 0 or below indicates they feel swaying. As seen in Figure 7, the mean value of the subjective stability score was lower than 0 only when participants wore rocker sole shoes through the rock road, which indicated that the participants felt physically unstable. This psychological characteristic suggests that in response to the effects of the rocker sole shoes and the rock road, subjects consciously increased motion control, in addition to instinctive motion control, to cope with the psychological perception of instability to maintain gait stability.

The correlation between the results of subjective stability scores and the ROM of joints was analyzed by Kendall’s tau-b correlation coefficient. The higher the correlation between the ROM of joints and the subjective stability rating, the higher the ROM of joints that causes the participants to perceive themselves as stable. As shown in Table 5, there is a correlation between the subjective stability scores and the ROM of joints for different footwear, and ground surfaces. The one with a strong positive correlation is the ROM of the hip joint in the coronal plane. It indicates that when the abduction–adduction of the hip joint becomes greater, correspondingly, participants perceive themselves to be more stable. Thus, the effect of footwear and ground surfaces can be psychologically established by increasing the ROM of the hip joint in the coronal plane. The results of the objective experiment showed that no significant difference was observed between the shoe and ground surfaces, in terms of the hip movement in the coronal plane, which means that the participants have not yet applied this lower limb adjustment to their daily life.

The ROM of the ankle also correlates with a correlation coefficient of 0.500. Objective experimental data suggest that the participants use the ankle movement for kinematic strategies, in response to the perceived instability associated with different footwear and ground surfaces. It indicates the importance of ankle joint movement to maintain gait stability. Future training in ankle joint activity could also be enhanced using a combination of footwear design and ground surfaces.

## 4. Discussion

### 4.1. Effects of Footwear and Ground Surface on ROM

In response to potential instability in footwear design, aging adults maintained gait stability through lower limb range of motion. Compared to young adults, aging adults adopted a more conservative kinematic strategy, mainly by reducing the ROM of the ankle joint. Compared with toe spring shoes, aging adults reduced the ROM of the ankle to a greater extent when wearing rocker sole shoes. Multivariate analysis of variance revealed no interaction between footwear and road factors in any lower limb range of motion. The ground surface also had a significant effect on the lower limb range of motion. Walking on grassland increased the ROM of the hip and knee joint (hip, *p* = 0.002; knee joint, *p* = 0.021) in the sagittal plane, increased the ROM of the ankle joint in the coronal plane, and decreased it in the transverse plane in aging adults, compared to level ground (*p* < 0.001). Walking on rock road decreased the ROM of the ankle joint in the transverse plane, compared to level ground (*p* < 0.001). Comparing rock roads, aging adults increased the ROM of the hip joint when passing over grassland (*p* = 0.042).

### 4.2. Effect of Footwear toe Spring and Rocker Sole Design on Kinematic Strategies

Regarding the effects of the toe spring design, the participants maintained gait stability by reducing the intrarotation–extrarotation of the ankle joint (level ground, *p* = 0.001; grassland, *p* = 0.006; rock road, *p* < 0.001). The literature indicated that the human body preferred to use the strategy of ankle joint dorsiflexion and knee joint extension when wearing toe spring shoes [41]. In addition, wearing toe spring shoes can reduce the resistance when the foot leaves the ground, and significantly decrease the stress on the forefoot in the later support phase [42].

When wearing rocker sole shoes, the participants mainly reduce the dorsiflexion–plantarflexion at the ankle joint and the flexion–extension at the knee joint in the sagittal plane (ankle joint: level ground, *p* = 0.008; grassland, *p* < 0.001; rock road, *p* = 0.002; knee joint: level ground, *p* = 0.003; grassland, *p* = 0.007; rock road, *p* = 0.004). The eversion–inversion at the ankle joint in the coronal plane (*p* = 0.048) was decreased when walking on the grassland. Additionally, the intrarotation–extrarotation at the ankle joint was reduced in the transverse plane (*p* < 0.001).

Comparing the difference in footwear toe spring design and rocker sole design, the participants have different joint movements in the sagittal plane under different ground surfaces. On level ground, the ROM of the knee joint (*p* = 0.004) were adjusted to maintain body balance. On the grassland, the movement of the ankle joint (*p* = 0.017) was controlled. The participants reduced the ROM of ankle joints in the transverse plane (level ground, *p* = 0.006; grassland, *p* = 0.021; rock road, *p* = 0.003) when wearing rocker sole shoes to maintain gait stability. The probable cause is that the rocker sole design brings greater instability to the wearer, compared to the toe spring design [36], which motivates the participant to adopt more conservative exercise methods.

### 4.3. Recommendations for Kinematic Control Strategies for Aging Adults

To cope with the potential safety risks with differences in footwear and ground surfaces, the aging adults will use kinematic control strategies to maintain gait stability; hence, their lower limb joint mobility should receive more attention. The function of rocker sole shoes as a balance training tool was confirmed in this study, since they can improve joint mobility. Different characteristics of ground surfaces can be combined with toe spring shoes and rocker sole shoes, and this information can be applied to gait stability training of aging adults. On the one hand, ground surfaces can be used to increase the impact on specific joint activities, and improve the training effect. For example, aging adults who pass over a rock road will reduce ankle joint movement in the transverse plane. Wearing rocker sole shoes will produce the same effects. Therefore, aging adults who need to improve transverse ankle joint movement control can consider wearing rocker sole shoes when walking over a rock road rather than level ground. On the other hand, the aging adults’ physiological function needs to be fully considered in such gait training. The literature mentioned that appropriate control was also necessary for intrarotation–extrarotation of the ankle joint, which may increase the risk of inward ankle joint sprains during walking [39].

## 5. Limitations

There are some limitations in this study. Only female participants were recruited; however, investigating the kinematic strategies of older males would also be meaningful. Additionally, the design of footwear involves many aspects, and this study only focused on the sole geometry. More footwear parameters will be investigated in future studies.

## 6. Conclusions

This research studies the influence of footwear and ground surfaces on the gait stability of aging adults. To ensure the safety of the experiment, the experiment introduced an age simulation suit, and conducted a feasibility analysis of the device. According to the kinematic strategies reflected in the kinematic parameters, and the subjective stability score results, the influences of footwear and ground surfaces on human gait stability were investigated.

In response to the potential instability factors in footwear design, the aging adults maintain a stable gait by adjusting their joint movements. A more conservative kinematic strategy, focusing on reducing the ROM of the ankle joint, compared with younger subjects, was adopted. Compared with the toe spring shoes, the ROM of the ankle joint when wearing rocker sole shoes decreased to a greater extent. Ground surfaces also have a significant impact on the angle of joint movement. When dealing with grassland and rock roads, the aging adults control the sagittal motion through the flexion and extension of the hip and knee joints. The eversion and inversion of the ankle joint in the coronal plane, and the intrarotation and extrarotation of the ankle joint in the transverse plane, were also controlled.

The findings of this study imply that rocker sole design can be used as a balance training tool to help improve gait stability in aging adults. Appropriately performing hip abduction and adduction can establish a sense of stability at the psychological level. Meanwhile, exercise training of the ankle joint in all planes can improve gait stability, and reduce the risk of falls in aging adults, which is beneficial for the sustainable well-being of the older adult population.

## Figures and Tables

**Figure 1 healthcare-10-02468-f001:**
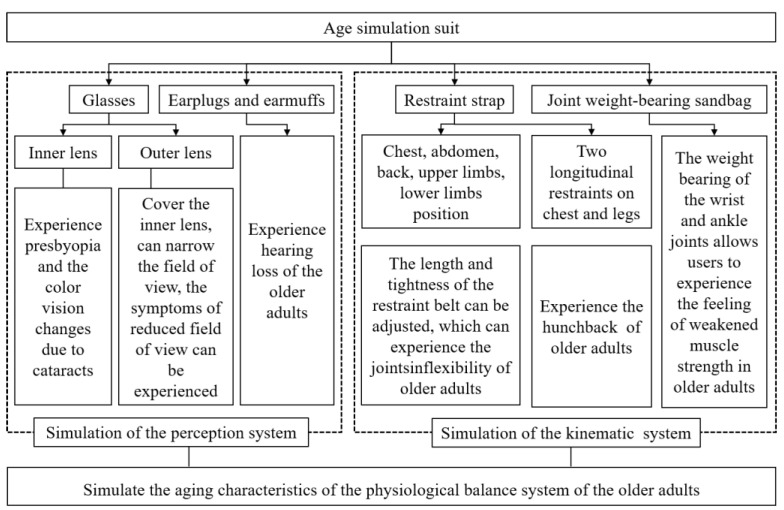
The principle of function realization for the age simulation suit.

**Figure 2 healthcare-10-02468-f002:**
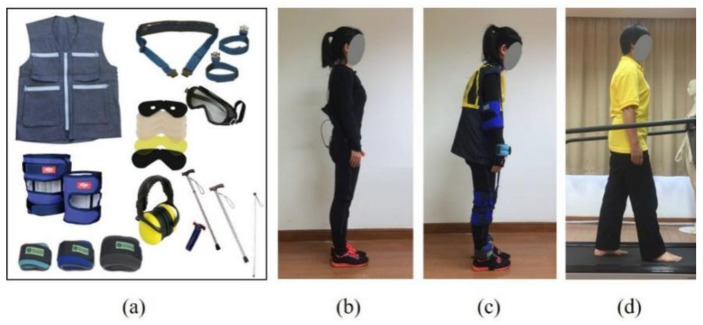
Age simulation suit and participants (**a**) age simulation suit; (**b**) younger adult; (**c**) younger adult with age simulation suit; (**d**) older adult.

**Figure 3 healthcare-10-02468-f003:**
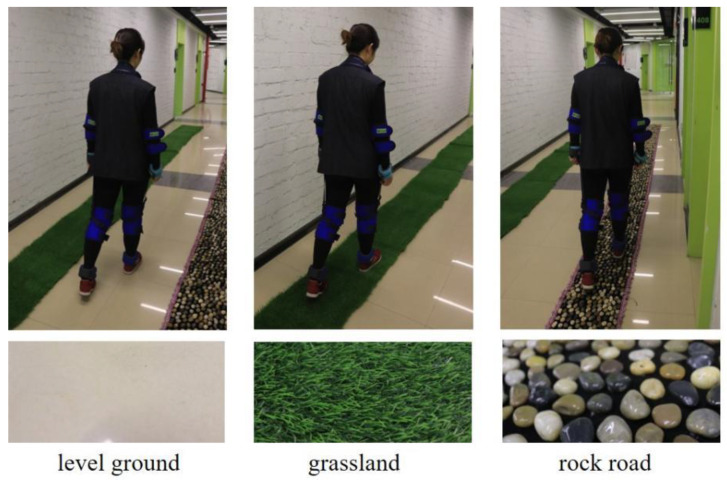
Ground surfaces with different characteristics.

**Figure 4 healthcare-10-02468-f004:**
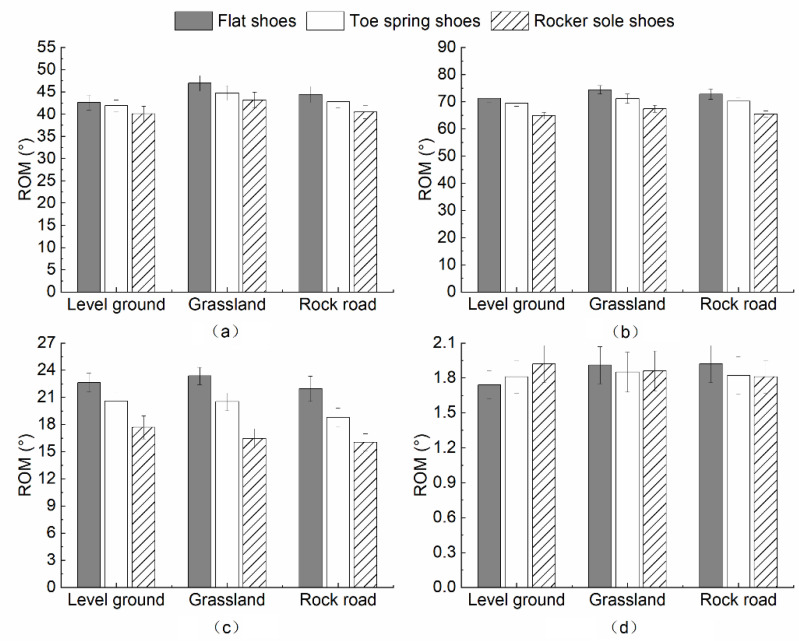
The influence of footwear on ROM in the sagittal plane under three ground surfaces (**a**) Hip; (**b**) Knee; (**c**) Ankle; (**d**) Ball of foot.

**Figure 5 healthcare-10-02468-f005:**
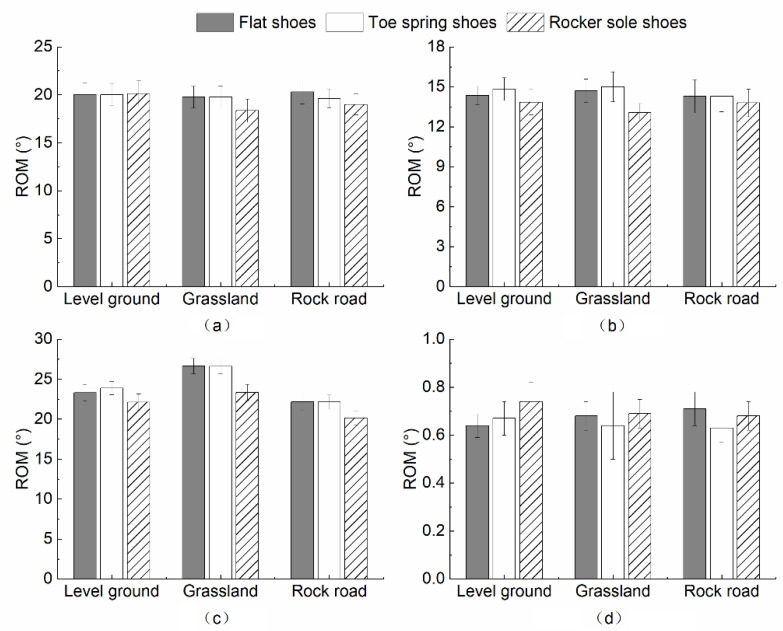
The influence of footwear on ROM in the coronal plane under three ground surfaces (**a**) Hip; (**b**) Knee; (**c**) Ankle; (**d**) Ball of foot.

**Figure 6 healthcare-10-02468-f006:**
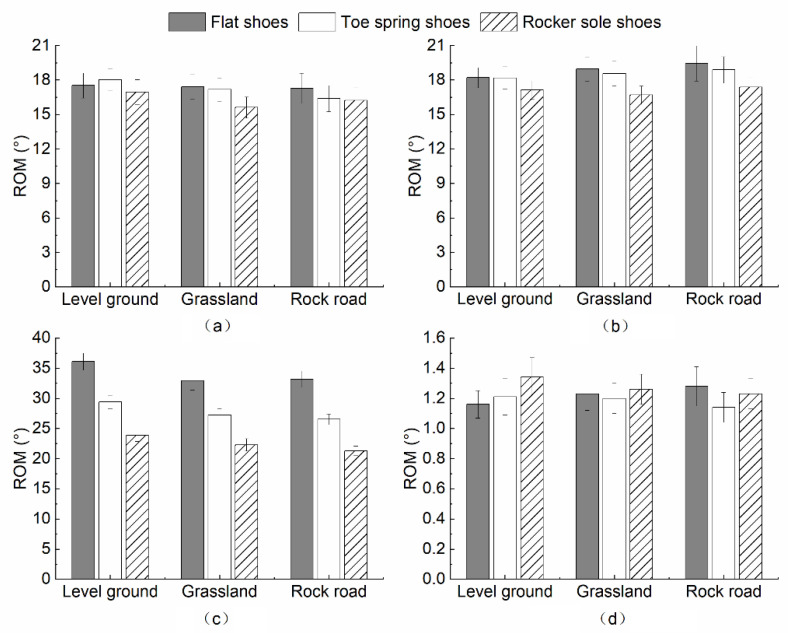
The influence of footwear on ROM in the transverse plane under three ground surfaces (**a**) Hip; (**b**) Knee; (**c**) Ankle; (**d**) Ball of foot.

**Figure 7 healthcare-10-02468-f007:**
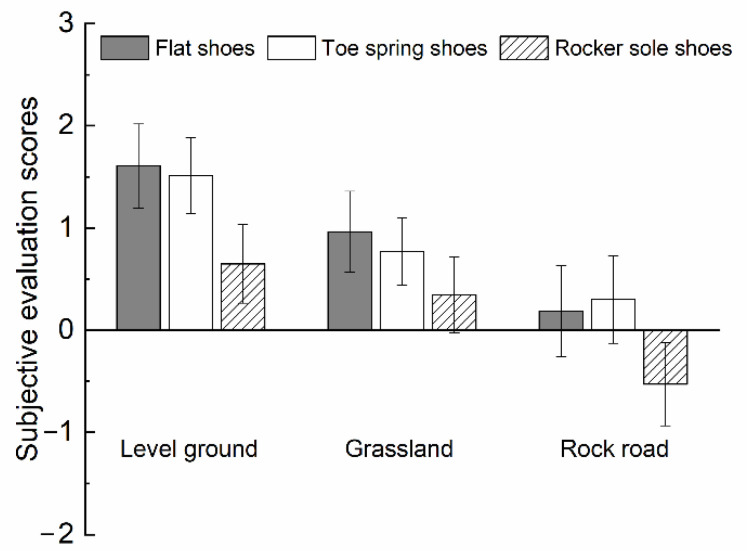
Effect of footwear on subjective human stability scores under three ground surfaces.

**Table 1 healthcare-10-02468-t001:** Experimental footwear.

Footwear	Side View	Sole Geometry	Sole Texture	Shoe Closure	Sole Material
The flat shoes	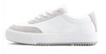	Flat sole	Horizontal stripes	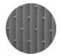	Shoelace	Rubber
The toe spring shoes	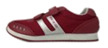	Toe spring	Horizontal twill	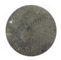	Velcro	Rubber
The rocker sole shoe	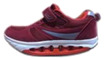	Rocker sole	Horizontal stripes	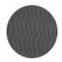	Velcro	Rubber

**Table 2 healthcare-10-02468-t002:** Statistical significance of ROM (*SD*) for the footwear and ground surfaces effects in the sagittal plane.

Ground Surface	Joint	ROM (°)	Flat Shoes	Toe Spring Shoes	Rocker Sole Shoes	Results of MANOVA			
						Statistical Quantity (F)	Degree of Freedom (*df*)		Significance (*p*)
Level ground	Hip	Est. M	42.60	41.89	40.09	0.688	2	39	0.508
		SD	6.16	4.82	6.41				
	Knee	Est. M	71.28	69.44	64.88	6.694	2	39	0.003 *
		SD	5.39	4.13	4.68				
	Ankle	Est. M	22.63	20.58	17.68	5.046	2	39	0.011 *
		SD	3.89	3.72	4.75				
	Ball of the foot	Est. M	1.74	1.81	1.92	0.384	2	39	0.684
		SD	0.45	0.54	0.60				
Grassland	Hip	Est. M	46.95	44.76	43.15	1.233	2	39	0.302
		SD	6.52	6.15	6.57				
	Knee	Est. M	74.42	71.16	67.42	5.264	2	39	0.009 *
		SD	5.74	6.40	4.90				
	Ankle	Est. M	23.35	20.52	16.46	12.123	2	39	<0.001 *
		SD	3.53	3.68	3.95				
	Ball of the foot	Est. M	1.91	1.85	1.86	0.040	2	39	0.961
		SD	0.60	0.62	0.62				
Rock road	Hip	Est. M	44.42	42.78	40.44	1.590	2	39	0.217
		SD	6.84	5.13	5.70				
	Knee	Est. M	72.78	70.21	65.49	5.994	2	39	0.005 *
		SD	7.07	5.15	4.38				
	Ankle	Est. M	21.94	18.77	16.04	6.769	2	39	0.003 *
		SD	5.16	3.92	3.46				
	Ball of the foot	Est. M	1.92	1.82	1.81	0.151	2	39	0.860
		SD	0.61	0.58	0.51				

Note: * Significance at *p* < 0.05, *SD* = Standard Deviation, Est. M = Estimated Mean.

**Table 3 healthcare-10-02468-t003:** Statistical significance of ROM (*SD*) for the footwear and ground surfaces effects in the coronal plane.

Ground Surface	Joint	ROM (°)	Flat Shoes	Toe Spring Shoes	Rocker Sole Shoes	Results of MANOVA			
						Statistical Quantity (F)	Degree of Freedom (*df*)		Significance (*p*)
Level ground	Hip	Est. M	20.02	20.01	20.12	0.002	2	39	0.998
		SD	4.59	4.28	5.07				
	Knee	Est. M	14.37	14.84	13.85	0.335	2	39	0.717
		SD	2.69	3.20	3.61				
	Ankle	Est. M	23.31	23.88	22.13	0.848	2	39	0.436
		SD	3.88	3.13	3.87				
	Ball of the foot	Est. M	0.64	0.67	0.74	0.559	2	39	0.576
		SD	0.20	0.27	0.29				
Grassland	Hip	Est. M	19.76	19.76	18.35	0.486	2	39	0.619
		SD	4.23	4.36	4.49				
	Knee	Est. M	14.71	15.00	13.08	1.310	2	39	0.281
		SD	3.25	4.20	2.52				
	Ankle	Est. M	26.63	26.59	23.33	3.828	2	39	0.030 *
		SD	3.65	3.40	3.80				
	Ball of the foot	Est. M	0.68	0.78	0.69	0.349	2	39	0.707
		SD	0.24	0.51	0.23				
Rock road	Hip	Est. M	20.30	19.62	19.00	0.346	2	39	0.710
		SD	4.65	3.60	4.06				
	Knee	Est. M	14.29	14.29	13.80	0.061	2	39	0.941
		SD	4.66	4.29	3.87				
	Ankle	Est. M	22.15	22.18	20.17	1.478	2	39	0.241
		SD	3.92	3.36	3.32				
	Ball of the foot	Est. M	0.71	0.63	0.68	0.371	2	39	0.692
		SD	0.27	0.22	0.22				

* Significance at *p* < 0.05. Note: *SD* = Standard Deviation, Est. M = Estimated Mean.

**Table 4 healthcare-10-02468-t004:** Statistical significance of ROM (*SD*) for the footwear and ground surfaces effects in the transverse plane.

Ground Surface	Joint	ROM/°	Flat Shoes	Toe Spring Shoes	Rocker Sole Shoes	Results of MANOVA			
						Statistical Quantity (F)	Degree of Freedom (*df*)		Significance (*p*)
Level ground	Hip	Est. M	17.52	18.04	16.95	0.266	2	39	0.768
		SD	4.07	3.70	4.05				
	Knee	Est. M	18.21	18.19	17.13	0.490	2	39	0.616
		SD	3.25	361	3.03				
	Ankle	Est. M	36.08	29.38	23.84	6.628	2	39	<0.001 *
		SD	5.26	4.21	3.73				
	Ball of the foot	Est. M	1.16	1.21	1.34	0.658	2	39	0.523
		SD	0.34	0.44	0.50				
Grassland	Hip	Est. M	17.41	17.17	15.62	0.936	2	39	0.401
		SD	4.02	3.82	3.43				
	Knee	Est. M	18.96	18.56	16.70	1.514	2	39	0.233
		SD	3.91	4.03	2.96				
	Ankle	Est. M	32.95	27.21	22.30	18.618	2	39	<0.001 *
		SD	5.86	3.92	3.80				
	Ball of the foot	Est. M	1.23	1.20	1.26	0.103	2	39	0.902
		SD	0.40	0.36	0.39				
Rock road	Hip	Est. M	17.28	16.39	16.23	0.230	2	39	0.796
		SD	4.86	4.28	4.08				
	Knee	Est. M	19.44	18.88	17.39	0.759	2	39	0.475
		SD	5.72	4.30	3.30				
	Ankle	Est. M	33.19	26.52	21.30	33.160	2	39	<0.001 *
		SD	5.00	3.29	3.03				
	Ball of the foot	Est. M	1.28	1.14	1.23	0.411	2	39	0.666
		SD	0.47	0.39	0.37				

Note: * Significance at *p* < 0.05, *SD* = Standard Deviation, Est. M = Estimated Mean.

**Table 5 healthcare-10-02468-t005:** Results of correlation analysis between subjective evaluation and movement parameters.

Movement Parameters		Correlation Coefficient	Significance of the Correlation Coefficient (*p*)
Sagittal plane	Hip	0.111	0.677
	Knee	0.222	0.404
	Ankle	0.500	0.061
	Ball of the foot	−0.222	0.404
Coronal plane	Hip	0.222	0.404
	Knee	0.444	0.095
	Ankle	0.500	0.061
	Ball of the foot	−0.111	0.677
Transverse plane	Hip	0.611	0.022 *
	Knee		1
	Ankle	0.500	0.061
	Ball of the foot	−0.278	0.297

Note: * indicates significant correlation at the 0.05 level.

## Data Availability

The datasets generated and analyzed during the current study are available from the author on reasonable request.

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
