# Peer review of "Kinematic Strategies for Sustainable Well-Being in Aging Adults Influenced by Footwear and Ground Surface"

_healthcare, 2022, doi:10.3390/healthcare10122468_

Round 1
Reviewer 1 Report
The manuscript is interesting. However, some key issues must be properly addressed in details:
1. Methodology. Page 3, L106-108. The age simulation suit by Sanwa Manufacturing Company is used in the study. If this suit design has been validated in any previous work? Any citations can be given?
2. Methodology. Page 5. L163. Any previous works to validate Xsens MVN in ROM measurements of lower limbs? It is suggested to give citations.
3. Methodology. Table 1. This study focuses on the footwear sole (geometry). It is suggested to reflect this key parameter and clarify in the title, results, discussion as well as conclusion.
4. Results. Page 7. L223-225. “Previous studies reported that …” However, only Lv’s study [36] is repeatedly cited throughout the results section. Any additional work can be referred? L228. “As the walking time increases…” However, subjects in this study only committed to a walkway of 7m at self-selected speed. The statement is not clear.
5. Results. Page 8. L278-280. Page 11. L346-347. It is suggested not using “>” to compare the changes at different footwear conditions.
6. Results. Page 10. L317-319. “Previous studies indicated that…” Again, only Lv’s study [36] was cited. Any other works can be cited?
7. Limitations of this study?
Author Response
The manuscript is interesting. However, some key issues must be properly addressed in details:
- Page 3, L106-108. The age simulation suit by Sanwa Manufacturing Company is used in the study. If this suit design has been validated in any previous work? Any citations can be given?
[Reply] Yes, we have validated the suit in our previous study, as seen in Reference 34.
- Page 5. L163. Any previous works to validate Xsens MVN in ROM measurements of lower limbs? It is suggested to give citations.
[Reply] Yes, the Xsens MVN has been used in many studies to measure the ROM of lower limbs. We also did some work with this equipment.
[1]Wang S, Wang Y. Musculoskeletal Model for Assessing Firefighters’ Internal Forces and Occupational Musculoskeletal Disorders During Self-Contained Breathing Apparatus Carriage[J]. Safety and Health at Work, 2022,13: 315-325.
- Table 1. This study focuses on the footwear sole (geometry). It is suggested to reflect this key parameter and clarify in the title, results, discussion as well as conclusion.
[Reply] Thanks for your suggestion. We have clarified this point in the title. In the main contents, we have defined flat shoes, toe spring shoes and rocker sole shoes to represent the differences of footwear sole.
- Page 7. L223-225. “Previous studies reported that …” However, only Lv’s study [36] is repeatedly cited throughout the results section. Any additional work can be referred? L228. “As the walking time increases…” However, subjects in this study only committed to a walkway of 7m at self-selected speed. The statement is not clear.
[Reply] Thanks for your comments. We have revised the sentence of L223-225, and L228.
- Page 8. L278-280. Page 11. L346-347. It is suggested not using “>” to compare the changes at different footwear conditions.
[Reply] Thanks for your suggestion. We have revised all related sentences.
- Page 10. L317-319. “Previous studies indicated that…” Again, only Lv’s study [36] was cited. Any other works can be cited?
[Reply] Thanks for your comments. We have revised the sentence.
- Limitations of this study?
[Reply] Thanks for your comments. We have added the limitation of this study.
Reviewer 2 Report
The fall is an inescapable problem influencing the health and threatening the safety of the older adults. Exploring the kinematic strategies of aging adults can help reduce the risk of falls.
The authors,to study kinematic strategies of aging adults in response to footwear (flat shoes, toe spring shoes, rocker sole shoes) and ground surfaces (level ground, grassland and rock road), a 3D motion capture system and subjective stability evaluation with 14 female participants were performed.
Their results indicated that footwear and ground surfaces significantly impacted joints dynamics during walking. Compared with young adults, aging adults tended to adopt a more conservative walking pattern. Wearing different shoes on the three ground surfaces mainly reduced the ROM (range of motion) of the ankle (p<0.05).
The authors said that rocker sole shoes gave aging adults a stronger sense of instability so they controlled the movement of ankle joint initiatively. When walking on grassland and rock road, aging adults adjusted the movements of hip, knee and ankle joints to maintain gait stability.
The authors concluded that aging adults are recommended to strengthen the flexibility training of the ankle joint, perform hip adduction and abduction exercises, and wear rocker sole shoes to improve their balance ability and sustainable well-being.
The study is innovative and well written.
I have a few comments with a pure academic spirit
1. In the abstract. Please explain the following text better “Rocker sole shoes gave aging adults a stronger sense of instability so they controlled the movement of ankle joint initiatively. When walking on grassland and rock road, aging adults adjusted the movements of hip, knee and ankle joints to maintain gait stability. The aging adults were recommended to strengthen the flexibility training of the ankle joint, perform hip adduction and abduction exercises, and wear rocker sole shoes to improve their balance ability and sustainable well-being.”
2. The purpose at the end of the introduction “The effects of footwear and ground surfaces on the changes in ROM of the lower limb in each plane were investigated, to explore the influence mechanism on gait stability.” Must be better expanded and explicated to valorize the valuable work.
3. Figure 2 must be described in details.
4. Introduce the paragraphs of the methods using a few sentences. This could help to understand the design of the study
5. Insert the limitations of the study in the discussion (all studies have limitations)
Author Response
- In the abstract. Please explain the following text better “Rocker sole shoes gave aging adults a stronger sense of instability so they controlled the movement of ankle joint initiatively. When walking on grassland and rock road, aging adults adjusted the movements of hip, knee and ankle joints to maintain gait stability. The aging adults were recommended to strengthen the flexibility training of the ankle joint, perform hip adduction and abduction exercises, and wear rocker sole shoes to improve their balance ability and sustainable well-being.”
[Reply] Thanks for your comments. This was concluded by analyzing both of objective and subjective evaluations. We have explained the conclusion in section 3.4 and 4.3. We also clarified this in the abstract.
- The purpose at the end of the introduction “The effects of footwear and ground surfaces on the changes in ROM of the lower limb in each plane were investigated, to explore the influence mechanism on gait stability.” Must be better expanded and explicated to valorize the valuable work.
[Reply] Thanks for your comments. We have expanded this point.
- Figure 2 must be described in details.
[Reply] Thanks for your suggestion. We have explained this figure in section 2.1.
- Introduce the paragraphs of the methods using a few sentences. This could help to understand the design of the study
[Reply] Thanks for your suggestion. We have explained the method at the beginning of the methodology.
- Insert the limitations of the study in the discussion (all studies have limitations)
[Reply] Thanks for your comments. We have added the limitation of this study.